# Sandwich (Amnion/Conjunctival-Limbal Autograft/Amnion) Transplantation for Recurrent Pterygium with Restrictive Strabismus

**DOI:** 10.3390/jcm11237193

**Published:** 2022-12-03

**Authors:** Hang Wong, Jia-Song Wang, Ya-Li Du, Hua-Tao Xie, Ming-Chang Zhang

**Affiliations:** Department of Ophthalmology, Union Hospital, Tongji Medical College, Huazhong University of Science and Technology, Wuhan 430022, China

**Keywords:** sandwich transplantation, recurrent pterygium, symblepharon, restrictive strabismus, esotropia, amniotic membrane transplantation, conjunctival limbal autograft

## Abstract

(1) Background: This study aimed to evaluate the clinical outcome of Sandwich (Amnion/Conjunctival-Limbal Autograft/Amnion) transplantation for recurrent pterygium with restrictive strabismus. (2) Methods: This retrospective study included 11 eyes in 11 patients diagnosed with recurrent pterygium with restrictive strabismus who received sandwich transplantation. The outcomes were measured by pterygium recurrence, best-corrected visual acuity, esotropia (prism diopters), and treatment complications. (3) Results: Eleven patients (six males, five females) had a mean age of 60.5 (range 36–80) years. The previously received pterygium excision surgery number was 1.8 ± 1.02 (range 1–4). The mean follow-up period was 19.9 ± 8.41 (range 12–36) months. All patients had a restriction of abduction in the previously operated eye, causing esotropia in the primary position. Pre-operative esotropia was 17.2 (range 10–30) prims diopter (PD). Five eyes (45.5%) had symblepharon before surgery. All patients were orthotropic until the last follow-up. Symblepharon was released in all eyes. Free ocular motility was present in all eyes. No donor site scar formation, scleral melt, or corneal ulcer was noted. (4) Conclusions: Sandwich transplantation for recurrent pterygium with restrictive strabismus is safe and effective.

## 1. Introduction

Recurrent pterygium is a challenging ocular surface disorder often resistant to conventional surgeries. It exhibits a more aggressive pattern, with a higher recurrence rate than primary pterygium [1]. A significant amount of conjunctival inflammation was caused by previous repeated surgeries. The scar formation was caused by inadvertent extraocular muscle injury during a previous procedure. Both lead to symblepharon, ocular motility restriction, and induce diplopia. These complications affected not only the cosmetic appearance but also the visual function [2,3,4].

Currently, the common treatment for recurrent pterygium associated with restrictive strabismus is to dissect the scar tissue following pterygium excision with tissue grafting, including conjunctival autografting (CAU) [2,5], conjunctival limbal autografting (CLAU) [6], amniotic membrane (AM) [7], and cryopreserved limbal allograft [8] with selective usage of adjuvant therapy, including anti-metabolic medications: mitomycin C (MMC) and 5-fluorouracil (5-FU) [8,9].

AM is widely used in ocular surface reconstruction [10,11]. Using the AM alone is not enough to halt recurrence in recurrent pterygium. Still, it is an ideal tissue graft, especially to cover the large conjunctival defect in recurrent pterygium [1]. CLAU is effective for recurrent pterygium [6], but the limited size of CLAU is not enough to cover the conjunctival defect. Anti-metabolic agents are widely used intraoperatively following recurrent pterygium removal, but the dosage and application time remains controversial [12,13]. It is associated with devastating complications, such as corneal edema, scleral melt, ulceration, and delayed conjunctival epithelialization [14,15,16].

We assumed that AM repaired the sizeable conjunctival defect and reformed the fornix, followed by CLAU and another AM covering the ocular surface, as the sandwich strategy, without any antineoplastic agents, is feasible for recurrent pterygium associated with restrictive strabismus. The purpose of this retrospective study was to review the clinical outcome of 11 consecutive eyes (11 patients) who underwent the Sandwich procedure.

## 2. Materials and Methods

This retrospective study was approved by the Ethics Committee of Wuhan Union Hospital according to the tenets of the Declaration of Helsinki (UHCT22652). All surgeries were performed by the same surgeon at the Wuhan Union Hospital.

This study consisted of consecutive patients diagnosed with recurrent pterygium with restrictive strabismus between August 2018 and August 2022. Restrictive strabismus was defined as the forced duction test positive. Patients were excluded if the primary diagnosis was pseudo-pterygium or the follow-up time was less than 12 months.

### 2.1. Data Analysis

Subject data were tabulated using Excel (Microsoft Office Excel 2016 MSO: Microsoft, Redmond, Wash). Frequencies of demographic and clinical variables were calculated. The data were expressed as the means ± SD and range.

### 2.2. Outcome Measurement

The outcome was measured in the following aspects: 1. Pterygium recurrence. Recurrence was defined as the presence of fibrovascular proliferative tissue crossing the limbus. 2. Best correct visual acuity. 3. Esotropia (prims diopter).

### 2.3. Preparation of Amniotic Membrane

The human amniotic membrane (AM) was prepared using the previously described method [17]. Briefly, after written informed consent, human placentas with umbilical cords were obtained immediately after elective cesarean section. Those positive for human immunodeficiency virus type 1 and 2, hepatitis B and C, and syphilis were excluded. After washing with saline, the AM was separated from the chorion by blunt dissection and then cut into 3 × 3 cm pieces. They were washed 3 times with saline containing 50 μg/mL penicillin, 50 μg/mL streptomycin, and 2.5 μg/mL amphotericin B before they were preserved in sterilized pure glycerin (Wuhan Union Hospital) at −20 °C for at most 3 months. Immediately before use, the membranes were thawed, glycerin was washed off by saline immediately, and immersed in saline for 10 min before use.

### 2.4. Surgical Procedure

An eyelid speculum was used to achieve adequate exposure (Figure 1A). Forced duction testing was performed to demonstrate the mechanical limitation of the extraocular movement (Figure 1A). Anesthesia was achieved using subconjunctival 2% lidocaine to balloon up the area of the pterygium and underlying Tenon’s fascia. The head of the pterygium was then peeled or scraped from the corneal surface with a surgical blade. The medial rectus muscle was exposed after the scar tissue removal (Figure 1B). The bare sclera was exposed following the complete resection of the fibrovascular tissue (Figure 1C). Then, a slightly oversized AM, aligned to the remained conjunctiva and underlying healthy Tenon, was secured to the sclera with an interrupted 8-0 absorbable suture densely (Figure 1D). CLAU was harvested from the superior limbus. Maintaining correct orientation, the limbal edge of the graft was aligned with the limbal edge of the pterygium site. The graft was fixed above the first layer of AM with interrupted 8-0 absorbable suture (Figure 1E, White Arrows). The second layer of a large AM covered the ocular surface entirely and was fixed with interrupted 8-0 absorbable suture (Figure 1F). 

### 2.5. Postoperative Management and Follow-Up

Tobradex (tobramycin 0.3% and dexamethasone 0.1%, Alcon, FortWorth, TX, USA) eyedrop was administered 4 times per day, followed by a weekly taper for 4 weeks thereafter, before shifting to 0.1% fluorometholone eye drops (Allergan, County Mayo, Ireland) 4 times per day followed by a weekly taper for 4 weeks. Preservative-free lubricants were used 4 times per day for 1 month. All patients were followed up daily for 1 week, weekly for 1 month, monthly for 3 months, and then at different intervals after surgery.

## 3. Results

### 3.1. Pretreatment Characteristics

The demographics of the study population are presented in Table 1. Six males (54.5%) and five females (45.5%) were included in this study. The patients’ mean age was 60.5 ± 11.9 (range, 36–80) years. The previous received pterygium excision surgery number was 1.8 ± 1.02 (range 1–4). Ocular motility restriction with limited abduction was seen in all patients, and symblepharon was present in five patients (45.5%). Pre-operation esotropia was 17.2 ± 5.8 (Range, 10–30) PD. Other diagnoses included seven cataracts (63.6%) and one IOL insertion (9%).

### 3.2. Recurrence

The mean follow-up period was 19.9 ± 8.41 (range 12–36) months. The first-quartile and third-quartile follow-up periods were 13.5 months and 25.5 months. No recurrence occurred in any of the eyes (Table 2). Symblepharon was released in all eyes, and no recurrence of symblepharon was observed in the last follow-up. Free ocular motility was present in all the eyes. No donor site scar formation, scleral melt, or corneal ulcer was noted.

### 3.3. Visual Acuity

At the last follow-up, 6 patients (54.5%) showed improvement in visual acuity, and five patients (45.5%) remained unchanged. No loss of vision was associated with the sandwich procedure (Table 2).

### 3.4. Esotropia

Five eyes (45.5%) were associated with symblepharon and inferior fornix obliteration before surgery. The reformation of the fornix was successful without the recurrence of scar formation. All patients were orthotropic in their primary gaze and gained full ocular motility at the last follow-up. The original data are documented in Appendix A.

### 3.5. Representative Case Reports

Case 2: A 36-year-old lady complained of right eye recurrent neoplasm and diplopia for 2 years. She had received four previous surgeries (one single excision, one excision with AM, and two excisions with CLAU) and was referred to our institution. Her visual acuity was 1.0. The ocular motility was restricted by the severe symblepharon and the scar tissue. (Figure 2A). Her esotropia was measured at 20 PD. After sandwich transplantation, no recurrence of pterygium and symblepharon occurred (Figure 2B), and her cosmetic appearance recovered perfectly at the 36-month follow-up (Figure 2C).

Case 4: A 60-year-old female complained of left eye restriction and decreased vision for 2 years. She had received three previous pterygium excisions before this referral. She was diagnosed with recurrent pterygium and restrictive strabismus. Ocular motility restriction in the abduction field was caused by the serve symblepharon (Figure 3A, White Arrow). Her esotropia was measured at 30 PD preoperatively. Her visual acuity was finger counting at 15 cm preoperatively since the scar tissue covered her central cornea (Figure 3B). Thirty-one months after sandwich transplantation, she recovered orthotropic (Figure 3C) with a smooth and stable ocular surface (Figure 3D). Her visual acuity on the last visit was 0.6.

## 4. Discussion

A single excision for the recurrent pterygium had over 80% recurrence rates [18,19]. CAU [7], CLAU [20,21], limbal allograft [22], AM [5,11,12], MMC [23], 5-FU [9], and Anti-VEGF injection [24] also proved to be effective in recurrent pterygium with varying recurrence rates Table 3. The combination use of the graft and the antineoplastic agent decreased the recurrence rate significantly [8,12]. MMC is a mainstream antineoplastic agent used in recurrent pterygium with an antifibrotic effect and was reported to be associated with devastating complications, including sclera necrosis and corneoscleral stromalysis [14,15,16]. Fallah, M. R. et al. also compared the transplantation of CLAU and AM vs. MMC and AM in the treatment of recurrent pterygium. CLAU with AM seems to be more effective than intraoperative MMC with AM for recurrent pterygium [25].

Recurrent pterygium is associated with more aggressive fibrovascular proliferation compared to primary pterygium [29,30]. The ocular surface inflammation caused by previous surgery induces symblepharon formation [8]. Moreover, the microtrauma of the medial rectus muscle during previous surgery causes fibrosis and scar formation [2,3]. Symblepharon and scar formation both lead to ocular motility restriction [2,3]. Five eyes in this study had symblepharon before surgery. The strategies for these cases should focus on preventing the recurrence of pterygium and the associated complications, including scar formation, symblepharon, restrictive strabismus, and ocular motility restriction, by controlling the inflammation and reconstructing the ocular surface.

In comparison to CAU, CLAU demonstrated better performance in reducing the recurrence rate of recurrent pterygium [20,21]. In these cases, with scar formation and symblepharon associated with inflamed ocular surfaces, a single use of CLAU after generous excision was insufficient to reconstruct the ocular surface. However, CLAU might create a barrier for the growth of subconjunctival fibrovascular tissue [31,32], and the limbal stem cells in CLAU provide epithelialization sources in ocular surface reconstruction [33]. The rapid re-epithelialization shortened the duration of the inflammation post-procedure, which might be related to this satisfying surgical outcome.

AM has been widely used in fornix reconstruction [10], ocular surface reconstruction [34], and recurrent pterygium for over 20 years [7,11,28]. The subconjunctival fibroblasts associated with Tenon’s fascia were considered to be the cells related to pterygium recurrence, and AM was proven capable of suppressing fibrosis in pterygium [35]. In recurrent pterygium with scar formation leading to restrictive strabismus, fibrotic tissue is extensively excised during the procedure and leaves a relatively sizeable conjunctival defect. AM has comparatively no size limitation and is an ideal graft to cover the conjunctival defect in these cases. The “Sealing the Gap” technique suggested by Tseng [36] was used in the first layer of AM, which covered the bare sclera and created a mechanical barrier against the truncated fibrovascular tissue in the present study.

In addition, the first layer of AM formed a basal surface for CLAU, provided the antifibrotic effect, and inhibited subconjunctival scarring. CLAU provided two cell sources: limbal and conjunctival progenitor cells for corneal and conjunctival epithelization. The second layer of large AM, which covered the entire ocular surface, aimed to stabilize the ocular surface, control the inflammation, and promote epithelialization. Two layers of AM maximized the antifibrotic and anti-inflammatory effects for halting the recurrence and recovering the non-inflamed ocular surface. 

The cosmetic need is one of the purposes for pterygium surgery. In addition to recurrent pterygium itself, symblepharon and restrictive strabismus aggravate the effect on cosmetic appearance. All eyes were orthotropic after sandwich transplantation and had a good cosmetic appearance. Restrictive strabismus also affects visual functions, which may induce diplopia and seriously depress the quality of life. No patients’ visual acuity decreased in the follow-up. Six eyes (54.5%) showed that the visual acuity improved, and five eyes (45.5%) remained unchanged. The limitation of this study is the small number of patients.

## 5. Conclusions

Sandwich transplantation is safe and effective in the treatment of recurrent pterygium with restrictive strabismus.

## Figures and Tables

**Figure 1 jcm-11-07193-f001:**
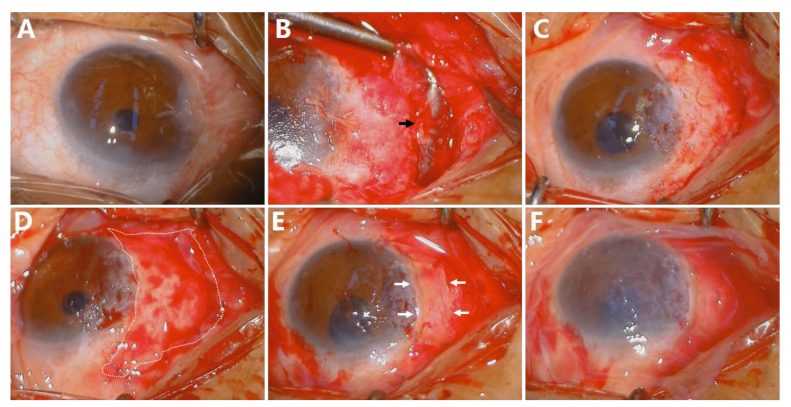
Surgical steps of Sandwich transplantation in recurrent pterygium with restrictive strabismus. Shows recurrent pterygium with restrictive strabismus pre-operation (**A**); Followed by dissection of pterygium. Medial rectus muscle was identified with a muscle hook (**Black Arrow**) (**B**); A sizeable conjunctival defect was remained after the removal of pterygium body and the scar tissue (**C**); The first layer of Amniotic membrane (white area) covered the bare sclera (**D**); CLAU was harvested, graft limbal side was aligned to the defect limbal side ((**E**), **White Arrows**); A large AM graft covered the ocular surface entirely (**F**).

**Figure 2 jcm-11-07193-f002:**
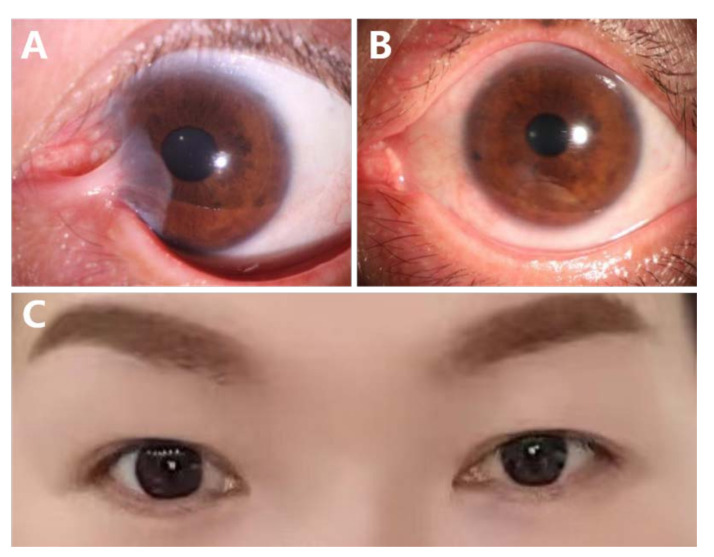
Case 2. The 36-year-old lady has recurrent pterygium, symblepharon, and limited abduction after four prior surgeries (**A**); No recurrence was observed after 36 months follow up with a silent ocular surface (**B**); Bilateral orthotropic and cosmetic appearance were restored (**C**).

**Figure 3 jcm-11-07193-f003:**
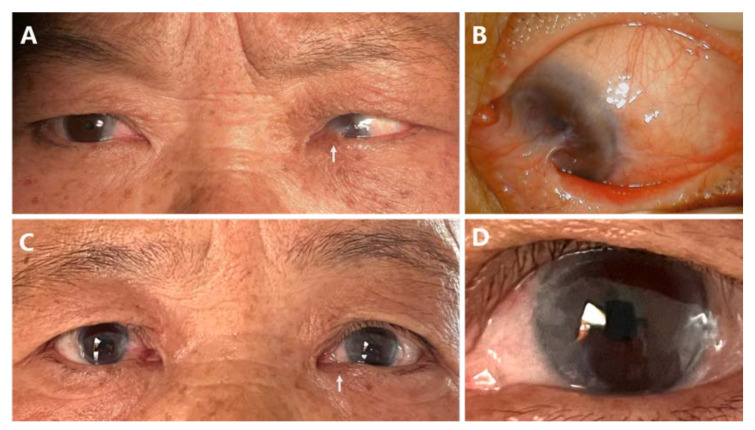
Case 4. A 60-year-old female underwent three prior pterygium surgeries with the sequela of serve symblepharon (white arrow) and total limitation of abduction (**A**). The scar tissue covered her central cornea (**B**). Thirty-one months after sandwich transplantation, she recovered orthotropic (**C**) with a smooth and stable ocular surface (**D**).

**Table 1 jcm-11-07193-t001:** Demographic and clinical features of patients with recurrent pterygium and restrictive strabismus who underwent the sandwich transplantation: pre-operation.

Demographic and Clinical Features	Data
Age (Year)	
Median	60
Mean ± SD	60.5 ± 11.9
Range	36–80
Gender	
Male	6 (54.5%)
Female	5 (45.5%)
Total prior pterygium surgeries	
Mean ± SD	1.8 ± 1.0
Range	1–4
Pre-operation Esotropia (PD)	
Mean ± SD	17.2 ± 5.8
Range	10–30
Fornix Obliteration	5 (45.5%)
Other diagnoses	
Cataract	7 (63.6%)
IOL insertion	1 (9%)

IOL = Intraocular lens.

**Table 2 jcm-11-07193-t002:** Recurrent pterygium with restrictive strabismus underwent sandwich transplantation: post-operation outcome.

Outcome	Data
Follow-up (Months)	
First quartile	13.5
Median (Second quartile)	16
Third quartile	25.5
Mean ± SD	19.9 ± 8.41
Range	12–36
Recurrence	
No	0 (0%)
Primary position	
Orthotropia	11 (100%)
Symblepharon	
No	0 (0%)
Best correct visual acuity	
Improved	6 (54.5%)
Unchanged	5 (45.5%)
Decreased	0 (0%)

**Table 3 jcm-11-07193-t003:** Previous reports of surgery for recurrent pterygium.

Author(s)	Year	Surgery	Recurrence Rate (%)	Number of Eyes	Follow Up Period (Months)	Minimum Follow Up (Months)
Kenyon et al. [5]	1985	CAU	7.3	41	22.6 ± 1.6	1
Riordan-Eva et al. [18]	1993	CAU	7	17	32	1
		Bare sclera	82	15	60	1
Tan et al. [19]	1997	CAU	0	17	13.2 ± 5.9	1
		Bare sclera	82	17		
Prabhasawat et al. [26]	1997	CAU	9.1	44	23.2 ± 20.3	3
		AM	37.5	8	0 ± 6.9	2.5
Al Fayez et al. [20]	2002	CAU	33.3	12	50	36
		CLAU	0	15	49	36
Nabawi et al. [27]	2003	CLAU	0	34	N/A	18
Fallah, M. R. et al. [25]	2008	CLAU + AM	0	20	N/A	6
		MMC + AM	20	20	N/A.	4
Shimazaki et al. [28]	1998	AM + CLAU	25	4	13.9 ± 6.4	3.7
Solomon et al. [1]	2001	AM	9.5	21	14.3 ± 4.9	6.3
Mashhoor et al. [21]	2013	CAU	10	112	63	36
		CLAU	1	112	61	36
Moden et al. [23]	2018	MMC + Double AM + Large CAU	0	31	43.2 ± 34.8	6
Moden et al. [8]	2020	MMC + Double AM + limbal allograft + CAU	0	10	36 ± 22.8	12

AM = amniotic membrane; CAU = conjunctival autograft; CLAU = conjunctival limbal autograft; MMC = mitomycin C; N/A = not available.

## Data Availability

The data used to support this study’s findings are available by contacting the corresponding author upon request.

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
