# Peer review of "Sandwich (Amnion/Conjunctival-Limbal Autograft/Amnion) Transplantation for Recurrent Pterygium with Restrictive Strabismus"

_jcm, 2022, doi:10.3390/jcm11237193_

Round 1

Reviewer 1 Report

The authors seem to present a promising surgical method for the treatment of recurrent pterygia.
Unfortunately, the text is not well written, so that - possibly due to linguistic deficiencies - it is often not clear what is meant. Therefore, the quality of the work cannot be judged sufficiently in its present state to make a recommendation for publication. The manuscript must be linguistically revised from the ground up, and a native speaker should do a final check.
Therefore: Check spelling and grammar, sentence structure, vocabulary, change between tenses (= is / was; upper/lower case), not publishable in this form.

Examples
e.g. line 11: e.g. - is to evaluated

Data analysis
Lines 63 - 64: unclear what is meant by "and range for quantitative parameters"

Page 3/4 Presentation of tables 1 and 2 in pdf incorrect.

Uniform notation AM or AMT (Table 4).

Are there any potential complications at the sampling site for CLAU?

Lines 154 - 161: are there references for these statements?

Line 162 - 168: unclear

Author Response

Point-to-Point Response to Reviewer 1

Dear reviewer,

Thank you for your constructive and kind comments. We have revised the manuscript carefully according to your comments and invited two native speakers for last review. We hope it is now suitable for publication on Journal of Clinical Medicine. Point-to-point responses are listed below.

Point 1:

  1. Examples e.g. line 11: e.g. - is to evaluated. Data analysis
  2. Lines 63 - 64: unclear what is meant by "and range for quantitative parameters" Page 3/4
  3. Presentation of tables 1 and 2 in pdf incorrect.
  4. Uniform notation AM or AMT (Table 3).

Response 1: Thanks sincerely for your reminder. We rewrote the sentences and deleted the unclear words in lines 63-64(Now in lines 71-72). We have rearranged the tables 1, 2 and 3 and checked the pdf.

Point 2: Are there any potential complications at the sampling site for CLAU?

Response 2: In this 11 cases study we did not observe scar formation and other obvious complications at the donor site. This result of donor site was added in result in lines 23 and 133.

Point 3: Lines 154 - 161: are there references for these statements?

Response 3: We have added more references to the statement in lines 154-161.

“Single excision for the recurrent pterygium had over 80% recurrence rate.

(Tan, D.T.H.; Chee, S.-P.; Dear, K.B.G.; Lim, A.S.M. Effect of Pterygium Morphology on Pterygium Recurrence in a Controlled Trial Comparing Conjunctival Autografting with Bare Sclera Excision. Archives of Ophthalmology 1997, 115, 1235-1240.

Riordan-Eva, P.; Kielhorn, I.; Ficker, L.A.; Steele, A.D.M.; Kirkness, C.M. Conjunctival autografting in the surgical management of pterygium. Eye 1993, 7, 634-638.)

Conjunctival autograft; (Kenyon, K.R.; Wagoner, M.D.; Hettinger, M.E. Conjunctival Autograft Transplantation for Advanced and Recurrent Pterygium. Ophthalmology 1985, 92, 1461-1470.)

conjunctival limbal autograft; (Mashhoor F, A.F. Limbal versus conjunctival autograft transplantation for advanced and recurrent pterygium. . Ophthalmology 2002, 109, 1752-1755.)

limbal allograft; (Ono, T.; Mori, Y.; Nejima, R.; Tokunaga, T.; Miyata, K.; Amano, S. Long-term follow-up of transplantation of preserved limbal allograft and amniotic membrane for recurrent pterygium. Graefe's archive for clinical and experimental ophthalmology = Albrecht von Graefes Archiv fur klinische und experimentelle Ophthalmologie 2016, 254, 2425-2430.),

AM; (Kenyon, K.R.; Wagoner, M.D.; Hettinger, M.E. Conjunctival Autograft Transplantation for Advanced and Recurrent Pterygium. Ophthalmology 1985, 92, 1461-1470)

MMC; (Tseng, S.C.; Prabhasawat, P.; Barton, K.; Gray, T.; Meller, D. Amniotic membrane transplantation with or without limbal allografts for corneal surface reconstruction in patients with limbal stem cell deficiency. Archives of ophthalmology (Chicago, Ill. : 1960) 1998, 116, 431-441.)

5-FU; (Pikkel, J.; Porges, Y.; Ophir, A. Halting pterygium recurrence by postoperative 5-fluorouracil. Cornea 2001, 20, 168-171.)

Anti-VEGF injection; (Lekhanont, K.; Patarakittam, T.; Thongphiew, P.; Suwan-apichon, O.; Hanutsaha, P. Randomized controlled trial of subconjunctival bevacizumab injection in impending recurrent pterygium: a pilot study. Cornea 2012, 31, 155-161.)

also shown effective in recurrent pterygium with varying recurrence rate [Table 3 in the article]. The combination use of the graft and the antineoplastic agent decreased the recurrence rate significantly. “

(Monden, Y.; Nagashima, C.; Yokote, N.; Hotokezaka, F.; Maeda, S.; Sasaki, K.; Yamakawa, R.; Yoshida, S. Management of Recurrent Pterygium with Severe Symblepharon Using Mitomycin C, Double Amniotic Membrane Transplantation, Cryopreserved Limbal Allograft, and a Conjunctival Flap. Int Med Case Rep J 2020, 13, 201-209.

Miyai T, H.R., Nejima R, Miyata K, Yonemura T, Amano S. Limbal allograft, amniotic membrane transplantation, and intraoperative mitomycin C for recurrent pterygium. Ophthalmology 2005, 112, 1263-1267.)

Point 4: Line 162 - 168: unclear

Response 4: We rewrote the paragraph in lines 162-168. The new paragraph was shown below:

“In comparison to CAU, CLAU demonstrated better performance in reducing the recurrence rate of recurrent pterygium. In these cases, with scar formation and symblepharon associated with inflamed ocular surfaces, a single use of CLAU after generous excision was insufficient to reconstruct the ocular surface. However, CLAU might create a barrier for the growth of subconjunctival fibrovascular tissue, and the limbal stem cells in CLAU provide epithelialization sources in ocular surface reconstruction. The rapid re-epithelialization shortens the duration of the inflammation post procedure, which may be related to this satisfying surgical outcome.”

Reviewer 2 Report

The work ““Sandwich” (Amnion/ Conjunctival-Limbal Autograft/ Amnion) Transplantation for Recurrent Pterygium with Restrictive Strabismus” is an interventional case series of 11 patients with recurrent pterygium and associated strabismus treated with a novel surgical technique which showed favorable outcomes. The field of recurrent pterygium constantly needs reevaluation, and this work is novel and important, exhibiting an elegant and promising technique, although limitations account for the small number of patients. The biggest limitation of this manuscript is its poor English. It needs substantial work including a full grammar review. I would suggest the author have a native English speaker or translator to revise the language and the writing fluency throughout the manuscript to improve its clarity, style, and grammar. Examples include (but are not limited to):

Line 29 : It was known as more

Line 40: Amniotic membrane (AM) is the innermost of the placenta

Line 49: We assume that apply AM to repair the large conjunctival defect and form the fornix

Line 66: The outcome measures as following aspect

Line 108: Pre-operation “mean +/- SD” Esotropia (capital letter)

Line 120: All patients were orthotropia

Other points:

This study consisted of consecutive patients diagnosed as recurrent pterygium with 58 restricted strabismus between August 2018 and May 2021. The authors should comment on how restricted strabismus was defined in the methods section. And also, that the number of recurrences was noted per patient.

Concerning patient selection: Was there a time window between surgery and the study initiation? Were patients excluded because of the follow up time? Or where only 11 patients treated with this surgery between August 2018 and May 2021? I see the range is from 12-53 months. Was 12 months the minimal timing for inclusion criteria? Because the follow up times are not normally distributed, the authors should write the median and the interquartile range. This will give the reader a better understanding of the actual timing of the study. Please clarify.

About the surgical technique, the authors did not use glue, correct? Was there any specific about the suture technique? Were these done in any specific way (continuous sutures vs interrupted?). Could you compare it to the “seal the gap” technique? (Cheng, A.M.S., Tseng, S.C.G. (2021). Sealing the Gap with Amniotic Membrane Transplantation for Primary and Recurrent Pterygium. In: Rosenberg, E.D., Nattis, A.S., Nattis, R.J. (eds) Operative Dictations in Ophthalmology. Springer, Cham. https://doi.org/10.1007/978-3-030-53058-7_4)

In the abstract, the authors mention that the study is prospective, yet in methods they mention is retrospective. Per my impression, is a retrospective interventional case series. Please clarify and modify in the manuscript.

Author Response

Point-to-Point Response to Reviewer 2

Dear reviewer,

Thank you for your constructive and kind comments. We have revised the manuscript carefully according to your comments and hope that it is now suitable for publication on Journal of Clinical Medicine. Point-to-point responses are listed below.

Point 1: This study consisted of consecutive patients diagnosed as recurrent pterygium with restricted strabismus between August 2018 and May 2021. The authors should comment on how restricted strabismus was defined in the methods section. And, that the number of recurrences was noted per patient.

Response 1: Thank you for your suggestion. We have added the definition of restrictive strabismus to the article in the methods section (Lines 65-66). The mean number of prior surgeries was list in Table 1.

The number of recurrences in each patient was documented in original data as Supplementary Table 1.

Point 2: Concerning patient selection: Was there a time window between surgery and the study initiation? Were patients excluded because of the follow up time? Or where only 11 patients treated with this surgery between August 2018 and May 2021? I see the range is from 12-53 months. Was 12 months the minimal timing for inclusion criteria? Because the follow up times are not normally distributed, the authors should write the median and the interquartile range. This will give the reader a better understanding of the actual timing of the study. Please clarify.

Response 2: Yes, there is a time window. This is a retrospective study. The duration is from August 2018 to August 2022. Patients who met the diagnostic criteria and were followed-up for at least 12 months were enrolled.

In the previous edition, May 2021 was the last operation time, not the follow-up time. We have corrected it. After revision, the median, first and third quartile of follow-up time is listed in table 2.

Point 3: About the surgical technique, the authors did not use glue, correct? Was there any specific about the suture technique? Were these done in any specific way (continuous sutures vs interrupted?). Could you compare it to the “seal the gap” technique? (Cheng, A.M.S., Tseng, S.C.G. (2021). Sealing the Gap with Amniotic Membrane Transplantation for Primary and Recurrent Pterygium. In: Rosenberg, E.D., Nattis, A.S., Nattis, R.J. (eds) Operative Dictations in Ophthalmology. Springer, Cham. https://doi.org/10.1007/978-3-030-53058-7_4)

Response 3: No glue was used in the procedure. Interrupted 8-0 absorbable suture was used to fix each layer of graft. Sealing the Gap technique was used in the first layer of AM fixation (for repairing the conjunctival defect). The remained conjunctiva and underlying healthy tenon was fixed by interrupted 8-0 absorbable suture very densely, secured to the sclera, and aligned to AM.

The CLAU also fixed with interrupted 8-0 absorbable suture. The second layer of AM cover the globe was fixed with interrupted 8-0 absorbable suture. Thanks for your suggestion again. We have added the suture methods and details in the surgical procedure section (lines 102-109).

Sandwich procedure is focused on the combination effect of the multi-graft. The first layer of AM covered the exposed muscle and bare sclera in the meantime. The second layer of AM covers the entire ocular surface to maximize the antiinflammation effect. The sandwich procedure not only repaired the nasal site of globe but also covered the CLAU harvested site. We added these in the discussion. (Lines 214-218)

Point 4: In the abstract, the authors mention that the study is prospective, yet in methods they mention is retrospective. Per my impression, is a retrospective interventional case series. Please clarify and modify in the manuscript.

Response 4: This is a retrospective case series. Thanks for pointing out our mistake. We have modified it in the abstract.

Round 2

Reviewer 1 Report

please check spelling and sentence structure again very carefully, verbs seem to be missing in some places. Otherwise the paper is now acceptable